# Inflammatory Pseudotumor of the Anal Canal Mimicking Colorectal Cancer: Case Report and Hints to Improve a Patient’s Fitness for Treatment and Prevention

**DOI:** 10.3390/diagnostics15070885

**Published:** 2025-04-01

**Authors:** Vito Rodolico, Paola Di Carlo, Girolamo Geraci, Giuseppina Capra, Cinzia Calà, Claudio Costantino, Maria Meli, Consolato M. Sergi

**Affiliations:** 1Department of Health Promotion, Mother and Child Care, Internal Medicine and Medical Specialties, University of Palermo, Piazza delle Cliniche 2, 90127 Palermo, Italy; paola.dicarlo@unipa.it (P.D.C.); giuseppina.capra@unipa.it (G.C.); cinzia.cala@unipa.it (C.C.); claudio.costantino@unipa.it (C.C.); maria.meli@unipa.it (M.M.); 2Department of Precision Medicine in the Medical, Surgical and Critical Care Areas, University of Palermo, Via Liborio Giuffrè 5, 90127 Palermo, Italy; girolamo.geraci@unipa.it; 3Department of Laboratory Medicine and Pathology, University of Alberta, Edmonton, AB T6G 2R3, Canada; 4Anatomic Pathology, Children’s Hospital of Eastern Ontario, University of Ottawa, Ottawa, ON K1H 8L1, Canada

**Keywords:** sexual behavior, anus neoplasms, sexually transmitted infections, HIV, HPV, endoscopy, inflammatory bowel disease mimicker, case report

## Abstract

**Background and Clinical Significance:** Men who engage in anal fisting may experience full rectal and colon thickness injury resulting in an endoscopic emergency. The endoscopist does not routinely question patients about their sexual habits, nor are patients compliant with counseling during the endoscopy procedure as indicated by the infectious disease clinician. **Case Presentation:** A 47-years-old HIV- and monkeypox virus (MPXV)-negative Caucasian gay man underwent colonoscopy because of changes in bowel habits with anal discomfort and rectal bleeding. The first colonoscopy showed a vegetative annular neoformation of the anal canal. There was a concentric stenosis of the lumen. The endoscopist suspected the diagnosis of anal squamous cell carcinoma and a histopathology investigation was requested. Biopsy histology excluded a frank neoplasm or anal intraepithelial neoplasia (AIN). Then, the patient was referred to a multidisciplinary team. With adequate counseling, the patient disclosed his habitual anal fisting. Laboratory identification of L1–L3 *Chlamydia trachomatis* (CT) genovars was positive for CT L1, L2, real-time PCR for *Neisseria gonorrhoeae* (NG), and *Mycoplasma hominis*. Human Papillomavirus (HPV)-DNA detection identified HPV type 70, 68, and 61. We illustrate this case with plenty of histology and immunohistochemistry. We also review the differential diagnosis of AIN according to the 5th edition (2019) WHO Classification of Digestive System Tumours. **Conclusions:** Our patient emphasizes two important aspects of endoscopy and pathology: first, the significance of understanding patients’ sexual behaviors in diagnosing rectal and colon injuries, as well as the need for sexually transmitted infections (STI) screening especially for CT; and second, the effectiveness of a multidisciplinary communication model that encourages private discussions to alleviate patients’ fears and improve prevention efforts.

## 1. Introduction

Sexually transmitted infections (STIs) pose a significant public health challenge. While many STIs are associated with traditional sexual practices, less common or unconventional practices also carry a risk of transmission, and due to their nature can cause trauma and lesions, increasing the vulnerability to infections [1]. Fisting, also known as handballing, fist-fucking, brachio-vaginal or brachio-proctic insertion, is an uncommon sexual practice consisting of the consensual or violent penetration of the vagina, anus, or both with the hand (fist) with or without the forearm [2,3]. During fisting sexual activity, several sexually transmitted infections (STIs) can be responsible for colon or rectal lesions [3,4,5,6,7]. According to the sexually transmitted infections (STI) treatment guidelines, an association exists with high-risk and traumatic sexual practices (e.g., condomless receptive anal intercourse or receptive fisting) and concurrent genital ulcerative disease or STI-related proctitis and sexually transmitted enteric infections [6]. Among STIs, Human Papillomavirus (HPV) is the most frequent worldwide and persistent infection, and its pathogenetic role in carcinogenesis is known both in women and men. The effects of non-intercourse receptive anal practices, such as fisting, on the transmission of anal high-risk HPV (HR-HPV) are currently being studied extensively [8,9].

This article reports a patient with an inflammatory pesudotumor exhibiting intense chronic and acute inflammatory changes in a setting of anal fisting (AF). It addresses the most common bacterial organisms that are sexually transmitted. Here, we also report poly-microbial identification in inflammatory lesions of the anorectal region after receptive fisting in an HIV-negative Caucasian subject who was referred to us for suspected anal cancer.

## 2. Case Report

In October 2023, a 47-year-old Caucasian male, active smoker, BMI 20.28, with high cultural status and negative relevant medical and surgical history, was referred by a family physician to undergo examination at the Endoscopy Unit of the academic public University Hospital “Paolo Giaccone” (A.U.O.P., University of Palermo, IT) for proctitis with severe tenesmus.

On examination, his temperature was 36.1 °C, blood pressure 110/70 mmHg, pulse 67 beats per minute, and oxygen saturation 96% when he was breathing ambient air. The endoscopy procedure was performed at the Endoscopy Unit of A.U.O.P. according to standardized hospital recommendation [10].

During physical examination, only warty lesions in the anal area were observed. He complained about changes in bowel habits with anal bleeding, pain, and sensation of a mass. The diagnosis on admission was of undetermined anal stenosis in the absence of anal warts. The interview on his history and habits was relevant for chronic constipation and his sexual practices within the context of homosexuality. The patient communicated no information about consensual rectal fisting intercourse. Common laboratory analysis and abdominal CT scan were normal, and before endoscopy procedure the patients showed recent negative investigation for Human Immunodeficiency Virus (HIV) and syphilis test performed at our academic public laboratory analysis. It appears that no HPV vaccination was administered or recommended by the family doctor. At colonoscopy, on the plane of the pelvic floor, there was a vegetative annular neoformation growing above the dentate line, provoking a remarkable stenosis of the lumen of the anal canal over 360°, with raised and fragile margins and ulcerated and fibrinous fundus (Figure 1a). The endoscopist suspected the diagnosis of anal squamous cell carcinoma. Multiple tissue biopsies were performed to establish the diagnosis. The histopathological examination revealed intense inflammatory granulation tissue with lymphocytic and plasmacytic infiltrates, with a prevalence of polytypic plasma cells associated with rare neutrophilic granulocytes. The external consultant (CMS) showed evidence of some mild erosive-ulcerative changes on the surface epithelium. There was no evidence of atypical epithelial elements (Figure 1b,c).

After the initial endoscopy, the patient did not return for further observation. However, after a month, the patient experienced additional persistent anal fullness and bleeding, prompting him to contact the endoscopist again. The physician then proceeded with a second investigation. Compared to the previous endoscopic examination, the mucosa of the distal rectum, starting from the last Houston fold (9 cm from the anal margin), appeared hyperemic and fragile. It was also noted to be edematous and easily bleeding spontaneously, with multiple reelevated rounded ulcerations at raised margins and fibrinous fundus. Biopsy sample confirmed the inflammatory changes.

Furthermore, concentric growth was found in the cranial direction of the new canal formation, annular in shape, vegetating and provoking a stenosis over 360° of the lumen of the anal canal (Figure 2a,b). Raised and fragile margins and fibrinous *fundus* were also present. Multiple biopsy sampling was repeated as well (Figure 2c).

After this second endoscopic procedure, the patient was referred to a multidisciplinary team service evaluation, which includes a pathologist with expertise in infectious disease, an infectious disease clinician, a clinical microbiologist, and a dermatologist. The patient, with adequate counseling, was asked about his consensual anal fisting intercourse with another man and confirmed that rectal bleeding had appeared immediately after anal fisting. Histological findings showed no evidence of reliable images of microorganisms using special stains and confirmed the clinical suspicion of condyloma, based on the finding of the anal mucosa characterized by images of koilocytosis, suggestive of HPV infection; p16 immunohistochemistry as a surrogate for transcriptionally active high-risk HPV given tested inconclusive result (Figure 2g and inset) with some areas of positive staining in both nuclear (focally) and cytoplasmic localization (mostly).

Following a second endoscopy, the patient was referred to the Infectious Diseases Unit at Policlinico “P. Giaccone” Hospital in Palermo, Italy. During the initial infectious disease consultation, throat, rectal, and urethral swabs, as well as urine samples, were collected for a Multiplex PCR test (Allplex™ STI Essential Assay, Seegene, Seoul, Republic of Korea) to detect seven sexually transmitted infections. Blood samples were also taken for point-of-care HIV and syphilis testing. Testing at the hospital’s Microbiology and Virology Unit revealed *Chlamydia trachomatis* (CT) L1-L2 serotypes (Viasure), *Neisseria gonorrhoeae* (NG), and *Mycoplasma hominis* through real-time PCR. Real-time PCR (Vircell-Microbiologists) for monkeypox virus (MPXV) was negative. HPV DNA testing (Allplex, HPV28 detection, Seegene) identified HPV types 70, 68, and 61 [11,12]. The patient received standard antibiotic treatment for the bacterial STIs, according to established guidelines, and initiated the HPV vaccination series [8].

After four weeks, multiplex PCR testing for STIs by PCR was negative and curative resection of the inflammatory pseudotumor of the anal canal by endoscopic submucosal dissection was offered to the patient.

After several months of attempting to reach the patient regarding missed endoscopy appointments and microbiological tests, we were informed by our Infectious Disease clinician that he had relocated to another country for professional opportunities.

Written informed consent was obtained initially from the patient to publish this paper; subsequently, since the patient left the city for traveling for work outside of the city boundaries, we also obtained informed consent from the family. In both events, we assured both patient and family that the case report will not disclose any data to avoid harm to either the patient or the family.

## 3. Discussion

Our case report highlights as some habits of anal sex, such as rectal fist insertion is an uncommon and potentially lethal dangerous sexual practice [1,2,3,4,5,13,14]. This practice is usually a homosexual activity, but it can also be a heterosexual or an autoerotic activity [5,15]. Recent studies encourage the enhancement of knowledge about the health status of lesbian, gay, bisexual, and transgender (LGBT) individuals, including sexual health, risk behaviors, kinky practices such as fisting, STIs, HIV rates, and relevant interventions to avoid stigma and promote protective factors for these populations [16,17,18]. Anal fisting may have some associated risks, which include rectal bleeding, either isolated or prolonged. More severe bleeding could indicate an underlying condition or injury requiring medical attention involving a multidisciplinary approach and confirmation of infection using laboratory tools to choose the most appropriate course of treatment and to screen at-risk groups, as seen in our patient [19,20,21]. In our report, the patient showed symptoms attributable to both anal fisting and related bacterial infections such as lymphogranuloma venereum (LGV). In our case report, a serovar L1–L2 of CT was identified, as reported in other European countries [19].

Recently anorectal manifestations of monkeypox are increasingly being recognized as a potentially serious complication [7,20].

Regarding the HPV genotype, more than one serotype may be found in squamous lesions exhibiting koilocytosis, which may suggest HPV infection as identified for cerivical intraepithelial neoplasia [21]. Our patient shows clearly that some sexual practices are responsible for the acquisition of multiple etiological agents’ infections. These microorganisms are causative of STIs, including HPV type 70 and 68, which are classified as probable/possible high-risk genotypes (HR-HPV). These genotypes have been recently reviewed (Table 1). The IARC monographs have highlighted significant evidence demonstrating that almost all persistent infections with specific HPVs can lead to cervical cancer [22], and our consultant (CMS) is a frequent chair for the preparation of IARC monographs in Lyon, France.

The detailed examination of this case raises the paramount consideration that it is essential to recognize that not all HPV types are equally carcinogenic. Some, such as HPV 16 and HPV 18, pose a substantial risk of cancer development, while others do not. Categorizing the least carcinogenic HPV types presents a challenge, but it is crucial for improving screening tests and advancing/supporting vaccine development/employment [19]. By identifying and targeting the most carcinogenic types, screening assays can enhance sensitivity and maintain specificity by excluding non-carcinogenic types. Furthermore, including the most carcinogenic types in multivalent HPV vaccines can help prevent a significant portion of cervical and other cancers. Therefore, the Working Group’s diligent efforts to evaluate the carcinogenic potential of each HPV type, focusing on evolutionary relationships and their presence in cases of invasive cancers, will continue to make valuable contributions to cancer prevention and control [20,21]. Since persistent infection with HR-HPV is a necessary cause for cancer development, DNA detection and genotyping of HPV can be an essential method for controlling and preventing HPV-related diseases [14]. HPV and CT are STIs with common transmission routes and risk factors. CT can promote HPV persistence and is associated with cervical cancer, but its co-infection and related clinical consequences are not well known. Understanding the prevalence, genotype distribution, and risk factors of asymptomatic CT and HPV infections is essential for effective prevention and interventions. P16INK4a immunohistochemistry is conventionally used to facilitate the diagnosis of HPV-associated cervical precancerous lesions. The investigation of positive p16 immunostaining signals is typically characterized by both nuclear and cytoplasmic staining, as shown in our positive control (Figure 2h). However, inconclusive or ambiguous results are a reality in p16 immunohistochemistry as well as in other areas of medicine. It is important to note that cytoplasmic staining alone should be considered negative. As previously reported, the p16 immunostaining pattern (clone E6H4, Ventana Medical Systems, Tucson, AZ, USA) combined with Ki67 (clone 30-9, Ventana Medical Systems, Tucson, AZ, USA) can be valuable in screening programs, especially when HPV infection persists. This approach also enhances our understanding of less common high-risk HPV genotypes [21,22]. As indicated, most p16 results are clearly positive or negative, but certain ones are and remain “ambiguous”, despite internal and external consultation. Such “ambiguous” diagnoses meet some but not all requirements for the “block-positive” pattern, as shown in Figure 2h. To the best of our knowledge, it is unclear whether ambiguous p16 immunoreactivity indicates oncogenic HPV infection or harbors a risk of progression, which remains a conundrum in gynecologic pathology [23,24]. In fact, Liu et al. studied extensively the p16 immunohistochemistry to classify cervical intraepithelial neoplasia 2 [25]. During 12-month surveillance, high-grade squamous intraepithelial lesions (HSILs) were detected in 35% of the p16 ”block-positive” group, 1.5% of negative group, and 16% of the “ambiguous group”. The accuracy of “ambiguous” p16 immunoreactivity in predicting oncogenic HPV and HSIL outcome is significantly lower than that of the block-positive pattern but greater than negative staining. The p16 immunoreactivity was based on four parameters: “(1) intensity: strong (dark brown color similar to the positive control) versus weak (yellow color significantly lighter than the positive control); (2) extent: diffuse (signal involves >50% of the epithelium) versus focal (<50% of the epithelium); (3) continuity: continuous (staining extends laterally over a significant distance) versus discontinuous (alternating clusters of either positively or negatively stained cells); and (4) location: positive cells reside in the lower third, two thirds, or full thickness of epithelium”. Thus, in agreement with the literature about “ambiguous” results, we would like to emphasize that specific guidelines for this intermediate category should be continuously reviewed to prevent diagnostic errors, because the immunohistochemical detection of p16 may in general practice not always be straightforward.

Recently, LGV should be considered as a differential in MSM presenting with proctitis or rectal mass, even without HIV co-infection [26].

CT-HPV coinfections represent also a challenge for physicians. Masiá et al. [27] showed co-infection with CT may potentiate the oncogenic capability of HPV16. It may increase the risk of high-grade anal intraepithelial neoplasia (HG-AIN) in people with HIV. HPV53 and HPV70 should be considered among the genotypes associated with HGAIN. It is worth noting that bacterial co-infections, including CT, *M. hominis*, and NG, as well as bacterial vaginosis-related species, can interact with HPV in the genital tract and lead to viral persistence and disease outcomes. However, it is essential to understand that co-infections involving HPV and diverse infectious agents also provide crucial insights into disease transmission and cancer progression, which can help in developing better treatment strategies [28]. In our clinical case, incomplete clinical information may lead to a delay in diagnosis. Two risk factors were present in our patient, including men having sex with men, and ano-receptive sex, which are also associated with AIN and squamous cell carcinoma.

AIN consists of non-invasive cytological and architectural dysplastic abnormalities of the squamous epithelium of the anal mucosa, usually associated with HPV infection. AIN was previously classified, according to low, moderate, and severe dysplasia, in grades 1, 2, and 3. The pathological evaluation of AIN has undergone multiple iterations and revisions over the years. The prevailing categorization, which is today widely acknowledged, is derived from the LAST project and categorizes the specimens into two distinct tiers [29]. The initial level is LSIL (low-grade squamous intraepithelial lesion), which incorporates AIN-1, while the subsequent level is HSIL (high-grade squamous intraepithelial lesion), which comprises AIN-2 and AIN-3. LSIL exhibits histologic features such as superficial cell atypia, characterized by enlarged nuclei and uneven nuclear membrane contours while maintaining the ratio of nucleus-to-cytoplasm. HSIL affects the entire epithelium or the lower two-thirds. It is characterized by a lack of maturation, increased darkness of the nucleus, irregularity of the membrane, and an increase in the ratio of nucleus to cytoplasm. Immunoperoxidase staining for p16 can help detect the presence of dark staining in the nucleus and cytoplasm in HSIL (Table 2).

The diagnosis of AIN is determined through the analysis of cellular samples or tissue samples obtained during routine tests. Anal cytology can be conducted at a primary care provider’s office using a swab. Suppose the screening results indicate the presence of a HSIL or LSIL-; in that case, individuals should be promptly referred for a formal biopsy and inflammatory bowel disease (IBD) or IBD-mimickers should also be considered [30,31,32,33]. Research investigating the sensitivity and specificity of anal cytology shows inconsistent results. The documented sensitivity of anal cytology in detecting any-grade AIN varies from 47–90%, with increased sensitivity for high-grade disease or substantially bigger affected areas. The specificity for anal cytology is rather low, ranging from 32% to 60%. Aside from cytology, a direct rectal examination is a crucial component of diagnosing AIN. Alterations in sphincter tone or abnormalities in the mucosa can suggest the presence of possible lesions that may require a biopsy. Although visual and digital rectal exams have advantages, they are inadequate for screening or diagnosing AIN. Conventional anoscopy or high-resolution anoscopy (HRA) can be used to perform a formal biopsy. This procedure usually produces enough tissue for microscopic examination to determine if LSIL or HSIL is present. Tissue biopsy offers the advantage of providing a greater understanding of the structure and, in certain situations, enable a more conclusive diagnosis when compared to cytology alone. The HRA is a valuable tool that may be used in the office to identify AIN and provide therapeutic intervention. Following the application of acetic acid, a magnifying anoscope is utilized to inspect the anus and lower rectum. Acetic acid enhances the visibility of dysplastic cells in relation to the surrounding tissue, aiding in the precise biopsy for pathological evaluation. A subsequent infectious disease microbiological evaluation, two-histopathology tests negative for neoplasia, and a thorough understanding of the patient’s sexual habits allowed the correct diagnostic classification and a clear improvement in the prognosis (Table 3). Anal fisting has been suggested as a non-violent mode of sexual and erotic practice between gay men. In consideration of the diffusion of this behavior, there is an urgent need for a comprehensive understanding of epidemiology, challenges, and emerging issues in bacterial STIs. Two of the four papers in The Lancet Regional Health—Europe Series on STIs help understand what exactly is on the rise, by delving into the epidemiology and the management of asymptomatic STIs in Europe [28]. STIs are a significant public health and social burden worldwide, especially in developing countries also related to challenges for privacy and confidentiality in the provision of care and use of services [34]. Laboratory tests are crucial to confirm STIs infections, select treatment, and screen high-risk groups to prevent anal cancer [3,4,5,6,7,8,28,34,35,36,37].

*M. genitalium* is an important rectal pathogen and is a cause of symptomatic proctitis. Clinical management of clinical features suggestive of proctitis should include testing for *M. genitalium* and its treatment. Moreover, gonococcal proctitis is a STI of the rectum caused by *Neisseria gonorrhoeae*. Anal itching, mucopurulent discharge, proctitis with rectal pain, constipation, and feeling a continual need to defecate, as well as rectal bleeding are common [38,39]. Finally, syphilis can cause proctitis, which can be quite pronounced. Syphilis is classically divided into three stages: primary (associated with a chancre or proctitis), secondary (associated with condyloma lata, skin rashes, or lymphadenopathy), or tertiary (associated with cardiac or gummatous lesions). If pseudotumors can be identified as pronounced degree of inflammation with stenosis of the anal canal, tumors apart from AIN can be represented by Kaposi sarcoma. In rare cases, infection with the human herpes virus 8 can lead to a Kaposi sarcoma (KS). Before HAART became widely used, the risk of KS in the general population was 20,000 times higher in the AIDS population, and as many as 21% of HIV positive MSM had KS. According to recent epidemiologic studies, the occurrence of KS has dropped significantly with the use of HAART. Although early signs of anorectal KS resemble hemorrhoids or other benign lesions, they are tiny, spherical, and purple in color. Biopsy is necessary for diagnosis confirmation. For most patients, intralesional radiation and chemotherapy can lead to a regression of the lesion as well as excellent cosmetic and palliative care. Only patients with severe or quickly developing cancer are considered for systemic treatment.

One kind of therapy that can help patients achieve their health goals is motivational interviewing (MI), which aims to strengthen the patient’s own internal drives to make positive behavioral changes. The MI process helps patients gain insight into their doubt about changing their behavior, helps them balance the merits and drawbacks of changing their behavior, and pushes them to achieve realistic and long-term goals. Working with LGBT-identified patients is made easier by MI’s patient-centric, nonjudgmental, and nonconfrontational attitude. When implemented properly, MI can assist people from historically oppressed groups become more resilient by placing more agency in their own hands [40,41]. Acceptance of LGBT patients is critical and should encompass four main areas: (1) worth, contrasting with a judgmental attitude and the practice of attaching conditions to value; (2) true empathy, delineating a practitioner as honest as he/she can be; (3) autonomy assistance, characterized by an approach how to accomplish the support without motivating the patients to modify their behavior, and (4) clear affirmation in an attempt to focus on the client’s positive qualities and achievements rather than pointing out what’s “wrong” with him/her. In this context, we would like to strengthen the importance of the patient-physician relationship in gathering more of the patient’s history using a contextualized discourse. In fact, an essential part of discourse analysis is identifying the societal and structural factors that shape a conversation’s context. For instance, a discussion in a bar would offer a very different setting than one in an office. By employing discourse contextualization approaches, scholars can decipher the literal meanings of words and their social connotations. These strategies consider several elements, such as the specific culture of the platform, user motives, and contextual clues. In sociolinguistics, contextualization indicates important details about an interaction or communication scenario using language, including both spoken and nonverbal cues [41,42,43,44]. This might provide a lot of information, like the people involved, the nature of their relationship, the location of the chat, and more. Andersen and Risør (2014) [41] found that these hints can be deduced from the speaker’s tone of voice, the type of language being used (formal versus informal), and how the language is utilized. Signs, both spoken and unspoken, that both speakers and listeners of a language employ to provide information about the context, setting, and nature of a discourse are known as contextualization cues [43]. The scholarly work of Basil Bernstein is an example of contextualization [42]. Bernstein explains how teaching settings, including textbooks, contextualize scientific information. The sole focus of contextualization within sociolinguistics is the examination of language use. The reason behind this is that sociolinguistics focuses on analyzing societal language use. To reiterate, contextualization signals are vital because they provide viewers with information that helps them make sense of the encounter.

Andersen and Risør (2014) [41] listed tone, accents, body language, language type, and facial expressions as some of the contextualization cues. Changing the pitch of a word is called intonation. A contextualized discourse can better identify emotions like interest, disappointment, or enthusiasm. One can learn a lot about a person’s values, cultural views, and place of origin, just by listening to their accent in a conversation. We can glean more information regarding the speaker’s relationship, emotions, and feelings towards the subject or other participants from facial expressions and body language [42,43,44,45,46,47]. The level of intimacy between two speakers can be discerned by their choice of formal or informal language. When done correctly, contextualization can significantly facilitate and improve the amount of information that can flow between a patient and his/her physician.

## 4. Conclusions

Overall, the presentation of our unusual lower gastrointestinal case is an educational instructive clinical vignette about an unconventional anal lesion (360° stenotic inflammatory pseudotumor) in an unusual setting. It discloses a successful multidisciplinary approach to patients with ano-rectal lesions. Taking a thorough clinical history may be challenging, and careful questioning on sex habits should always ensure client privacy to prevent identification and reduce stigmatization. This physician’s commitment to confidentiality and privacy encourages continued access to services and motivates potential clients to seek help. There are a few key points that may be suggested: (1) Vaccination camping is available and stressful in young adults who are male and have risky sexual behaviors; (2) sexual behaviors or habits involve the emotional-affective sphere, and the patient is not always available at the first visit to talk about his/her sexual habits, considering that these practices are not only widespread in the youth or cases of sexual violence; (3) as the incidence of STIs continues to rise, it would be beneficial for surgeons and clinicians to include data on protection usage in their medical assessments; (4) in surgical setting, the endoscopist is the first doctor who is confronted with the patient who practices AF. The surgeon must be alerted to the possible multiple infections that a subject can acquire during fisting practices that affect not only HIV infection and syphilis but also other STIs; and (5) future research on the role of other non-intercourse receptive anal sexual practices in LGBT subjects, such as AF, in anal HR-HPV transmission or reactivation should be explored.

## Figures and Tables

**Figure 1 diagnostics-15-00885-f001:**
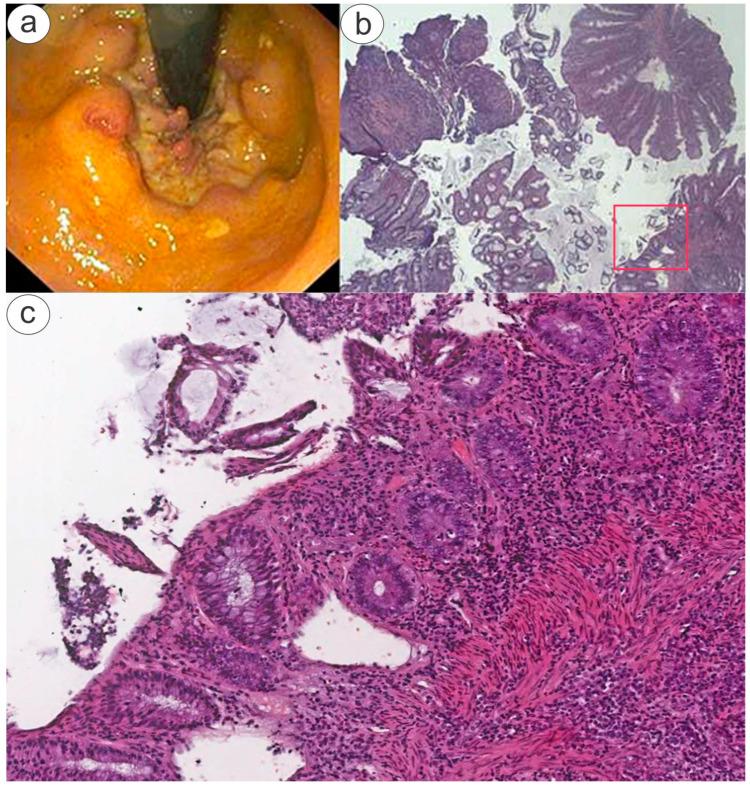
First colonoscopy: (**a**) There is an excavated lesion of the rectum (retroversion). (**b**) First biopsy sampling of mucosa of the rectum: intense lymphocyte infiltrates are evident also in the submucosa layer, with a prevalence of plasma cells associated with rare neutrophilic granulocytes as well as unclear evidence of erosive-ulcerative lesions on the surface epithelium (**c**) High-power inset arising from the red square depicted in (**b**), ×40 original magnification). No evidence of atypical epithelial elements that could support the clinical suspicion of “anal squamous cell carcinoma” (H&E stain; ×2.5 original magnification).

**Figure 2 diagnostics-15-00885-f002:**
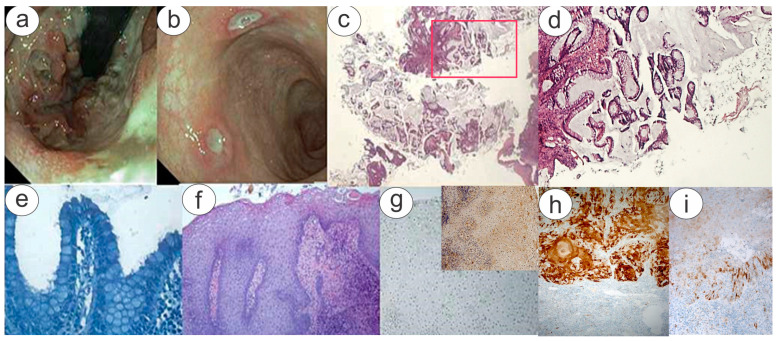
Second colonoscopy: There is hyperemia and fragility of the mucosa of the distal rectum (retroversion, (**a**)), with multiple elevated rounded ulcerations at raised margins and fibrinous fundus (**b**). (**c**,**d**) Second biopsy sampling of mucosa of the distal rectum: fragments of glandular mucosa in abundant mucous lake, associated with leukocyte-fibrin clots (**c**) (H&E stain; ×2.5 original magnification, (**d**), high-magnification inset from the rectangle depicted in (**c**) at ×20 original magnification). The Giemsa stain (**e**) did not reveal structures referable to microorganisms on the surface of the glandular lumens or at the intracellular site (Giemsa stain; ×40 original magnification). The epithelium of the anal mucosa, comprised in sampling, showed acanthosis and papillomatosis with images of koilocytosis, suggestive of HPV infection (**f**) H&E stain; ×20 original magnification); p16INK4a immunohistochemistry as a surrogate for transcriptionally active high-risk HPV given tested an ambiguous result with focal nuclear and cytoplasmic staining (**g**) and high magnification (inset)) (Immunohistochemical analysis of paraffin-embedded tissue using p16INK4a Antibody followed with DAB staining; ×10 original magnification). (**h**) Positive control run at the same time of staining the patient’s specimen and using the same antibody and (**i**) negative control run at the same time of staining the patient’s specimen and using the same antibody (clone E6H4, Ventana Medical Systems, Italy).

**Table 1 diagnostics-15-00885-t001:** High-risk (HR) genotype and low-risk (LR) HPV genotype according to the International Agency for Research on Cancer (IARC) classification.

HR HPV Genotypes	LR HPV Genotypes
16	6
18	11
26 *	40
31	42
33	43
35	44
39	54
45	55
51	61
52	62
53	64
56	71
58	72
59	81
66	83
67	84
68a	87
68b	89
69	90
70	
73	
82	

Note: * indicates 26th genotype is marked with a special note. From: https://www.cancer-environnement.fr/fiches/publications-du-circ/classification-du-circ-par-localisations-cancereuses/ (accessed on 24 March 2025).

**Table 2 diagnostics-15-00885-t002:** Terminology applied to various anal dysplastic lesions and p16 features on immunohistochemistry as a surrogate for transcriptionally active high-risk HPV.

Description	AIN	LAST	P16 IHC
Mild dysplasia	AIN 1	Low grade SIL	Negative or weak staining
Moderate dysplasia	AIN 2	High grade SIL	Positive and sometimes patchy staining
Severe dysplasia	AIN 3	High grade SIL	Diffuse, strong staining; nuclear and cytoplasmic “block-like”
Carcinoma in situ	AIN 3	High grade SIL	Diffuse, strong staining; nuclear and cytoplasmic “block-like”

(Adapted from WHO Classification of Tumours—Digestive System Tumours—Anal squamous dysplasia (intraepithelial neoplasia)—5th ed. 2019) [29]. AIN, anal intraepithelial neoplasia; IHC, immunohistochemistry; LAST, Lower Anogenital Squamous Terminology (LAST) project.

**Table 3 diagnostics-15-00885-t003:** Screening tests for anal high-grade squamous intraepithelial lesion (HSIL) and cancer.

Screening Test	Triage Test	LEV	Special Considerations
**Cytology**	None HR-HPV **(±genotyping)**	BII	Anal cytology is the most widely used and evaluated test for anal cancer screening.
		CII	HR-HPV testing to triage ASC-US cytology could be used to reduce HR referral rates.
**HR-HPV** **(±genotyping)**	None	BII	The efficiency of primary testing with a pooled HR-HPV test is limited in populations with high HPV prevalence (e.g., MSM with HIV). This strategy is useful in settings with no cytological infrastructure, or to reduce HRA (for HR-HPV negative patients) in practices providing HRA on all patients. Additional triage may be needed. Use of HR-HPV genotyping, specifically for HPV16, may help identify patients with high risk of HSIL or cancer. Performance does not seem to improve with the addition of HPV18.
	Cytology	CII	HRHPV testing to triage ASC-US cytology could be used to reduce HRA referral rates.
**HR-HPV** **(±genotyping)**	None	BII	The efficiency of primary testing with a pooled HR-HPV test is limited in populations with high HPV prevalence (e.g., MSM with HIV). This strategy could be considered in settings with no cytological infrastructure, or to reduce HRA (in HR-HPV negative patients) in practices providing HRA on all patients. In most settings, additional triage will be needed for HR-HPV positive individuals. Use of HR-HPV genotyping may help identify patients with high risk of HSIL or cancer.
**Cytology/HR-HPV co-test** **(±genotyping)**	None	BII	Anal co-testing does not provide any benefit over primary HR-HPV testing for anal HSIL. Anal co-testing may be especially beneficial for its negative predictive value. Co-testing may be less efficient in populations with high HR-HPV prevalence.
**DARE**	None	BII	All populations at-risk for anal cancer receive DARE at time of screening tests (or in lieu of screening tests in absence of HRA availability).

(Adapted from WHO Classification of Tumours—Digestive System Tumours—Anal squamous dysplasia (intraepithelial neoplasia)-5th ed. 2019) [29]. Abbreviations: ASC-US, atypical squamous cells of undetermined significance; DARE, digital anal rectal examination; HR, high-risk; HRA, high-resolution anoscopy; HSIL, high-grade squamous intraepithelial lesion; LEV, level of evidence; MSM, men who have sex with men. Adapted from Source: [37].

## Data Availability

The original contributions presented in this study are included in the article. Further inquiries can be directed to the corresponding authors.

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
