# Peer review of "Inflammatory Pseudotumor of the Anal Canal Mimicking Colorectal Cancer: Case Report and Hints to Improve a Patient’s Fitness for Treatment and Prevention"

_diagnostics, 2025, doi:10.3390/diagnostics15070885_

Round 1
Reviewer 1 Report (Previous Reviewer 2)
Comments and Suggestions for Authors
Review Report
Title: Multidisciplinary Approach to address Multiple Sexual Infections in a Patient with Inflammatory Pseudotumor of the Anal Canal Mimicking Colon Rectal Cancer after Habitual Anal Fisting. Case Report and Hints to improve a Patient’s Fitness for Treatment and Prevention
Authors: Vito Rodolico, Paola Di Carlo, Girolamo Geraci, Giuseppina Capra, Cinzia Calà, Claudio Costantino, Maria Meli , and Consolato M. Sergi
Comments:
In their case report “Multidisciplinary Approach to address Multiple Sexual Infections in a Patient with Inflammatory Pseudotumor of the Anal Canal Mimicking Colon Rectal Cancer after Habitual Anal Fisting”. Rodolico and colleagues reports a patient with an inflammatory pesudotumor exhibiting intense chronic and acute inflammatory changes in a setting of Anal Fisting.
Taken together, the study is adds substantial new knowledge regarding unconventional anal lesion and discloses a successful multidisciplinary approach to patients with ano-rectal lesions.
The authors respond well for my previous review report and carried out the suggested modifications.
The authors should, however add some minor additional modifications to the manuscript:
- The authors changed the title of their report. It becomes too long!
It is better to remove this part from the title “Case Report and Hints to improve a Patient’s Fitness for Treatment and Prevention”
- Also, in the title the authors wrote (Colon Rectal Cancer), it is better to be written colorectal cancer.
- In line 26: it is better for the reader to write MPXV is abbreviation for what! Please add monkeypox virus before MPXV
- In line 62: The authors wrote AF for the first time in the report as abbreviation. I/t is better for the reader to add the words (Anal Fisting) before AF for the first time in the report.
- In figure 2 (c), Pleas make the red rectangle a little bit larger to include exactly all the tissue area shown in (d
- In Figure 2 (h); Positive controls are used to confirm that your IHC staining of a specimen is working as expected.
- In the author’s case, Positive control should be staining tissue known to be positive for p16INK4a immunohistochemistry at the same time of staining the Patient’s specimen and using the same antibody. It is not correct to be online image for p16INK4a immunohistochemistry and say this is the Positive control!
It is essential to run Positive and negative controls in IHC staining experiments to confirm that the observed staining pattern is accurate and reliable.
- In line 121: Please correct the name of the stain. The authors wrote (H&H stain). It should be (H&E). it is abbreviation for Hematoxylin &Eosin.
- In line 126: Please correct the name of the stain. The authors wrote (E&E stain). It should be (H&E). it is abbreviation for Hematoxylin &Eosin.
Author Response
In their case report “Multidisciplinary Approach to address Multiple Sexual Infections in a Patient with Inflammatory Pseudotumor of the Anal Canal Mimicking Colon Rectal Cancer after Habitual Anal Fisting”. Rodolico and colleagues reports a patient with an inflammatory pesudotumor exhibiting intense chronic and acute inflammatory changes in a setting of Anal Fisting.
Taken together, the study is adds substantial new knowledge regarding unconventional anal lesion and discloses a successful multidisciplinary approach to patients with ano-rectal lesions.
The authors respond well for my previous review report and carried out the suggested modifications.
Thank you for your comments and suggestions. We changed the manuscript according to your suggestions.
The authors should, however add some minor additional modifications to the manuscript:
- The authors changed the title of their report. It becomes too long!
It is better to remove this part from the title “Case Report and Hints to improve a Patient’s Fitness for Treatment and Prevention”
Thank you for your comments and suggestions. We shortened the manuscript's title.
- Also, in the title the authors wrote (Colon Rectal Cancer), it is better to be written colorectal cancer.
Yes, indeed. We changed it.
- In line 26: it is better for the reader to write MPXV is abbreviation for what! Please add monkeypox virus before MPXV
Yes, indeed. Thank you.
- In line 62: The authors wrote AF for the first time in the report as abbreviation. I/t is better for the reader to add the words (Anal Fisting) before AF for the first time in the report.
Thank you. We changed it.
- In figure 2 (c), Please make the red rectangle a little bit larger to include exactly all the tissue area shown in (d
Yes, we changed accordingly.
- In Figure 2 (h); Positive controls are used to confirm that your IHC staining of a specimen is working as expected.
- In the author’s case, Positive control should be staining tissue known to be positive for p16INK4a immunohistochemistry at the same time of staining the Patient’s specimen and using the same antibody. It is not correct to be online image for p16INK4a immunohistochemistry and say this is the Positive control!
It is essential to run Positive and negative controls in IHC staining experiments to confirm that the observed staining pattern is accurate and reliable.
Yes, we changed accordingly. We retrieved the original positive and negative control slides run together with the specimen of the case report and we re-composed the photograph.
- In line 121: Please correct the name of the stain. The authors wrote (H&H stain). It should be (H&E). it is abbreviation for Hematoxylin &Eosin.
Done, thank you.
- In line 126: Please correct the name of the stain. The authors wrote (E&E stain). It should be (H&E). it is abbreviation for Hematoxylin &Eosin.
Done, thank you.

Reviewer 2 Report (New Reviewer)
Comments and Suggestions for Authors
This insightful case report highlights the critical importance of early detection through STI screening for patients with tumor-like lesions that resemble anal or rectal cancer. This issue is particularly pertinent to men who have sex with men (MSM) and those who engage in anal fisting. There is a well-documented link between sexual activity and the development of anal cancer in both MSM and heterosexual individuals. Recognized sexual risk factors for anal cancer include receptive anal intercourse, having multiple sexual partners throughout one’s lifetime, and the presence of sexually transmitted infections (STIs). While human papillomavirus (HPV) and human immunodeficiency virus (HIV) are the STIs most commonly associated with anal cancer, other STIs that have been observed include anal or genital warts, herpes simplex virus type 2 (HSV-2), Chlamydia trachomatis, gonorrhea, and syphilis.
The manuscript includes important figures, and the tables provide valuable information. The abstract is well-written; however, there is limited discussion on sexually transmitted infections (STIs) and their impact on the development of rectal tumors and stenosis. The authors fail to address any roles of Gonorrhea and Mycoplasma infections in HIV transmission and the development of tumor-like lesions that resemble anal or rectal cancer.
On a positive note, the detailed discussion of anal intraepithelial neoplasia (AIN), anal cytology, high-resolution anoscopy, HPV genotypes, and the use of p16 immunochemistry as a surrogate marker for transcriptionally active high-risk HPV is excellent.
It is important to standardize terminology throughout the text; for instance, "STI" should be consistently used instead of "STD", (line 38).The discussion section of the manuscript requires revision, and few paragraphs should be rewritten for clarity. Additionally, the complete references included in the discussion section (specifically lines 240–241 and 242–247) should be retained in the reference list only, not in the discussion section.
Furthermore, the discussion on the patient-physician relationship and motivational interviewing is overly lengthy and should be made more concise and straightforward.
It is essential to ensure that the conclusion is clear, concise, and aligned with the conclusions presented in the abstract. This will help summarize the key points and maintain reader engagement.
Author Response
This insightful case report highlights the critical importance of early detection through STI screening for patients with tumor-like lesions that resemble anal or rectal cancer. This issue is particularly pertinent to men who have sex with men (MSM) and those who engage in anal fisting. There is a well-documented link between sexual activity and the development of anal cancer in both MSM and heterosexual individuals. Recognized sexual risk factors for anal cancer include receptive anal intercourse, having multiple sexual partners throughout one’s lifetime, and the presence of sexually transmitted infections (STIs). While human papillomavirus (HPV) and human immunodeficiency virus (HIV) are the STIs most commonly associated with anal cancer, other STIs that have been observed include anal or genital warts, herpes simplex virus type 2 (HSV-2), Chlamydia trachomatis, gonorrhea, and syphilis.
Thank you for your comments and suggestions. We added details on these microorganisms potentially giving rise to proctitis, and occasionally prominent proctitis with pseudotumor-features.
The manuscript includes important figures, and the tables provide valuable information. The abstract is well-written; however, there is limited discussion on sexually transmitted infections (STIs) and their impact on the development of rectal tumors and stenosis. The authors fail to address any roles of Gonorrhea and Mycoplasma infections in HIV transmission and the development of tumor-like lesions that resemble anal or rectal cancer.
On a positive note, the detailed discussion of anal intraepithelial neoplasia (AIN), anal cytology, high-resolution anoscopy, HPV genotypes, and the use of p16 immunochemistry as a surrogate marker for transcriptionally active high-risk HPV is excellent.
Thank you for your comments and suggestions. We changed the manuscript accordingly.
It is important to standardize terminology throughout the text; for instance, "STI" should be consistently used instead of "STD", (line 38).The discussion section of the manuscript requires revision, and few paragraphs should be rewritten for clarity. Additionally, the complete references included in the discussion section (specifically lines 240–241 and 242–247) should be retained in the reference list only, not in the discussion section.
Yes, we revised the discussion and changed STI for STD.
Furthermore, the discussion on the patient-physician relationship and motivational interviewing is overly lengthy and should be made more concise and straightforward.
This section was shortened.
It is essential to ensure that the conclusion is clear, concise, and aligned with the conclusions presented in the abstract. This will help summarize the key points and maintain reader engagement.
We changed according and the conclusion is shortened.

This manuscript is a resubmission of an earlier submission. The following is a list of the peer review reports and author responses from that submission.
Round 1
Reviewer 1 Report
Comments and Suggestions for Authors
The article presented, offers an important perspective on the risks of sexual activities. The authors address the topic in an appropriate and comprehensive manner. The need for careful collection of the medical history and the risks of drop-out during follow-up are rightly emphasized. I am not convinced about the appropriateness of identifying the risk associated with fisting practices solely within the male population. In fact, there are bisexual orientations, and although to a lesser extent, this issue is also present in the female population. The diverse LGBTQIA+ community should be considered in order to avoid minimizing this problem, which is perhaps under-represented in the literature more due to the limited knowledge within the medical field than to delayed patient presentation. Moreover: HIV-positive gay, HIV-positive bisexual men and men who engage in receptive anal intercourse with a higher number of recent sexual partners have an elevated risk of incident anal high-risk human papillomavirus infection; non-intercourse receptive anal practices are not independently associated with incident anal high-risk human papillomavirus infection.
Comments on the Quality of English LanguageMinor editing of English language required.
Author Response
The article presented, offers an important perspective on the risks of sexual activities. The authors address the topic in an appropriate and comprehensive manner. The need for careful collection of the medical history and the risks of drop-out during follow-up are rightly emphasized. I am not convinced about the appropriateness of identifying the risk associated with fisting practices solely within the male population. In fact, there are bisexual orientations, and although to a lesser extent, this issue is also present in the female population. The diverse LGBTQIA+ community should be considered in order to avoid minimizing this problem, which is perhaps under-represented in the literature more due to the limited knowledge within the medical field than to delayed patient presentation. Moreover: HIV-positive gay, HIV-positive bisexual men and men who engage in receptive anal intercourse with a higher number of recent sexual partners have an elevated risk of incident anal high-risk human papillomavirus infection; non intercourse receptive anal practices are not independently associated with incident anal high-risk human papillomavirus infection.
At a time when are an increasingly open, acknowledged, and visible part of society, clinicians and researchers are faced with incomplete information about the health status of this community.
[Reply]: Thank you very much for your comments and suggestions. We have revised the manuscript accordingly. The following sentences and references were added in the Introduction, discussion and conclusion section.
Introduction section
An association exists with high-risk and traumatic sexual practices (e.g., condomless receptive anal intercourse or receptive fisting) and concurrent genital ulcerative disease or STI-related proctitis and sexually transmitted enteric infections
The effects of non-intercourse receptive anal practices, such as fisting, on the transmission of anal High-Risk HPV (HRHPV) are currently being studied.
Twisk, D. E., B, M. A., Van Eeden, A., Heideman, D. A., M, F. R., De Vries, H. J., & F, M. (2018). Detection of Incident Anal High-Risk Human Papillomavirus DNA in Men Who Have Sex With Men: Incidence or Reactivation? The Journal of Infectious Diseases, 218(7), 1018-1026. https://doi.org/10.1093/infdis/jiy276
Wong IKJ, Poynten IM, Cornall A, Templeton DJ, Molano M, Garland SM, Fairley CK, Law C, Hillman RJ, Polizzotto MN, Grulich AE, Jin F; SPANC study team. Sexual behaviours associated with incident high-risk anal human papillomavirus among gay and bisexual men. Sex Transm Infect. 2022 Mar;98(2):101-107. doi: 10.1136/sextrans-2020-054851. Epub 2021 Mar 16. PMID: 33727339; PMCID: PMC8862078.
Discussion section
Recent studies encourage enhanced knowledge about the health status of lesbian, gay, bisexual, and transgender (LGBT) individuals including sexual health, risk behaviors, kinky practice such as fisting, STIs, HIV rates, and relevant interventions to avoid stigma and promote protective factors for these populations.
Bitzer, Johannes. "Sexual and Reproductive Healthcare for LGBTI." Textbook of Contraception, Sexual and Reproductive Health (2024): 347.
Pavanello Decaro, S., Pessina, R., Biella, M., & Prunas, A. (2024). Italian women who have sex with women: Prevalence and co-occurrence of sexual practices. Sexual Medicine, 12(2). https://doi.org/10.1093/sexmed/qfae017
Conclusion
Future research on the role of other non-intercourse receptive anal sexual practices in LGBT subjects, such as fisting, in anal High Risk HPV transmission or reactivation could explored.
Comments on the Quality of English language: Minor editing of English language required.
Thank you again!

Reviewer 2 Report
Comments and Suggestions for Authors
Review Report
Title: Annular Neoformation of the Anal Canal Mimicking Neoplasia after Habitual Anal Fisting in a Man Showing Carcinogenic 3 HPV Genotypes and Differential Diagnosis with Anal Intraepithelial Neoplasia
Authors: Vito Rodolico, Paola Di Carlo, Girolamo Geraci, Giuseppina Capra1, Cinzia Calà, Claudio Costantino, Maria Meli and Consolato M. Sergi
Comments:
In their case report “Annular Neoformation of the Anal Canal Mimicking Neoplasia after Habitual Anal Fisting in a Man Showing Carcinogenic 3 HPV Genotypes and Differential Diagnosis with Anal Intraepithelial Neoplasia”.
The study is well designed, demonstrating a short medical story highlighting uncommon habit of anal sex, such as rectal fist insertion which is potentially lethal dangerous sexual practice. Notably, the authors present the case in a clear way, They highlight uncommon problem , may be misinterpreted as neoplasia. They gave detailed history about the patient then showed the correlation between endoscopic examination and histopathological examination of biopsies.
Taken together, the study is well conducted and adds substantial new knowledge to the field of sexually transmitted infections (STIs) when diagnosing and treating injuries caused by sexual behavior; which can be interpreted as either anal neoplasia or an inflammatory bowel disease mimicker.
The authors should, however add some minor additional informations the manuscript:
In line 7, There are numbers 3,4 above the author named “Consolato M. Sergi”, but in the following lines, there is no affiliation corresponding to number 4!
Figure 1 (b): it is better to add also a higher magnification for H&E image showing what is written in the figure legend. Most of the written histologic features cannot be seen clearly from this very low magnification X 2.5. It is better to add a higher magnification for a descriptive area in this figure.
Figure 2 (C): it will be better to use a higher magnification than x2.5 to show the morphological features in a clear way.
In line 99, correct E&E to be H&E stain.
In line 168 and 169, The two sentences are not clear enough to the reader. The authors need to rewrite them with adding the name of P16 immunostaining to confirm positivity of HPV.

Author Response
In their case report “Annular Neoformation of the Anal Canal Mimicking Neoplasia after Habitual Anal Fisting in a Man Showing Carcinogenic 3 HPV Genotypes and Differential Diagnosis with Anal Intraepithelial Neoplasia”.
The study is well designed, demonstrating a short medical story highlighting uncommon habit of anal sex, such as rectal fist insertion which is potentially lethal dangerous sexual practice. Notably, the authors present the case in a clear way, They highlight uncommon problem , may be misinterpreted as neoplasia. They gave detailed history about the patient then showed the correlation between endoscopic examination and histopathological examination of biopsies.
Taken together, the study is well conducted and adds substantial new knowledge to the field of sexually transmitted infections (STIs) when diagnosing and treating injuries caused by sexual behavior; which can be interpreted as either anal neoplasia or an inflammatory bowel disease mimicker.
The authors should, however add some minor additional informations the manuscript:
In line 7, There are numbers 3,4 above the author named “Consolato M. Sergi”, but in the following lines, there is no affiliation corresponding to number 4!
[Reply]: Thank you for your suggestion. The change is made
Figure 1 (b): it is better to add also a higher magnification for H&E image showing what is written in the figure legend. Most of the written histologic features cannot be seen clearly from this very low magnification X 2.5. It is better to add a higher magnification for a descriptive area in this figure.
[Reply]: an inset (x40) has been added
All figures are presented with higher resolution and TIF files are also uploaded.
Figure 2 (C): it will be better to use a higher magnification than x2.5 to show the morphological features in a clear way.
[Reply]: an inset (x20) has been added
In line 99, correct E&E to be H&E stain.
[Reply]: Thank you for your suggestion. The change is made.
In line 168 and 169, The two sentences are not clear enough to the reader. The authors need to rewrite them with adding the name of P16 immunostaining to confirm positivity of HPV.
[Reply]: Thank you for your suggestion. The sentences were modified and the reference changed
P16INK4a immunohistochemistry is conventionally used to facilitate the diagnosis of HPV-associated cervical precancerous lesions The investigation of positive p16 immunostaining signals is typically characterized by both nuclear and cytoplasmic staining, as shown in our positive control (Figure 2h). However, inconclusive or ambiguous results are a realty in p16 immunohistochemistry as well as in other areas of medicine. It is important to note that cytoplasmic staining alone should be considered negative. As previously reported, the p16 immunostaining pattern (clone E6H4, Ventana Medical Systems, Tucson, AZ, USA) combined with Ki67 (clone 30-9, Ventana Medical Systems, Tucson, AZ, USA) can be valuable in screening programs, especially when HPV infection persists. This approach also enhances our understanding of less common high-risk HPV genotypes [20].. As indicated, most p16 results are clearly positive or negative, but certain ones are and remain “ambiguous”, despite internal and external consultation. Such “ambiguous” diagnoses meet some but not all requirements for the “block-positive” pattern, as shown in Figure 2h. To the best of our knowledge, it is unclear whether ambiguous p16 immunoreactivity indicates oncogenic HPV infection or harbors a risk of progression (REF: Liu Y, Alqatari M, Sultan K, Ye F, Gao D, Sigel K, Zhang D, Kalir T. Using p16 immunohistochemistry to classify morphologic cervical intraepithelial neoplasia 2: correlation of ambiguous staining patterns with HPV subtypes and clinical outcome. Hum Pathol. 2017 Aug;66:144-151. doi: 10.1016/j.humpath.2017.06.014. Epub 2017 Jul 11. PMID: 28705710; PMCID: PMC5644341). In fact, Liu et al. studied extensively the p16 immunohistochemistry to classify cervical intraepithelial neoplasia 2 (Liu Y, Alqatari M, Sultan K, Ye F, Gao D, Sigel K, Zhang D, Kalir T. Using p16 immunohistochemistry to classify morphologic cervical intraepithelial neoplasia 2: correlation of ambiguous staining patterns with HPV subtypes and clinical outcome. Hum Pathol. 2017 Aug;66:144-151. doi: 10.1016/j.humpath.2017.06.014. Epub 2017 Jul 11. PMID: 28705710; PMCID: PMC5644341). During 12-month surveillance, high-grade squamous intraepithelial lesions (HSILs) were detected in 35% of the p16” block-positive” group, 1.5% of “negative group”, and 16% of the “ambiguous group”. The accuracy of “ambiguous” p16 immunoreactivity in predicting oncogenic HPV and HSIL outcome is significantly lower than that of the block-positive pattern but greater than negative staining. The p16 immunoreactivity was based on four parameters: “(1) intensity: strong (dark brown color similar to the positive control) versus weak (yellow color significantly lighter than the positive control); (2) extent: diffuse (signal involves > 50% of the epithelium) versus focal (< 50% of the epithelium); (3) continuity: continuous (staining extends laterally over a significant distance) versus discontinuous (alternating clusters of either positively or negatively stained cells); and (4) location: positive cells reside in the lower third, two thirds, or full thickness of epithelium”. Thus, in agreement with the literature about “ambiguous” results, we would like to emphasize that specific guidelines for this intermediate category should be continuously reviewed to prevent diagnostic errors, because the immunohistochemical detection of p16 may in general practice not be straightforward at all times.
Also, a reference was added:
Cabibi, D., Napolitano, C., Giannone, A. G., Micciulla, M. C., Porcasi, R., Lo Coco, R., Bosco, L., Vinciguerra, M., & Capra, G. (2021). Predictive Role of the p16 Immunostaining Pattern in Atypical Cervical Biopsies with Less Common High Risk HPV Genotypes. Diagnostics, 11(11), 1947. https://doi.org/10.3390/diagnostics11111947

Reviewer 3 Report
Comments and Suggestions for Authors
I read with interest the manuscript by Rodolico et al. titled “Annular Neoformation of the Anal Canal Mimicking Neoplasia after Habitual Anal Fisting in a Man Showing Carcinogenic HPV Genotypes and Differential Diagnosis with Anal Intraepithelial Neoplasia.” The manuscript explores the important topic of anal tumors, with intriguing insights from a challenging case report. Please find my comments below:
- Please disclose the final diagnosis clearly in both the abstract and the main manuscript to enhance clarity for the readership.
- I recommend that the authors follow validated guidelines for reporting case reports, such as the SCARE guidelines, to ensure comprehensive and standardized reporting.
- It is unclear why the authors delve into the differential diagnosis of AIN when an AIN diagnosis was ultimately excluded in this case. This point requires clarification or removal to maintain focus.
- Please elaborate on why this case is unique and novel compared to the existing literature. Highlight the aspects that make it a significant contribution to the topic.
Minor English editing
Author Response
I read with interest the manuscript by Rodolico et al. titled “Annular Neoformation of the Anal Canal Mimicking Neoplasia after Habitual Anal Fisting in a Man Showing Carcinogenic HPV Genotypes and Differential Diagnosis with Anal Intraepithelial Neoplasia.” The manuscript explores the important topic of anal tumors, with intriguing insights from a challenging case report. Please find my comments below:
Please disclose the final diagnosis clearly in both the abstract and the main manuscript to enhance clarity for the readership.
I recommend that the authors follow validated guidelines for reporting case reports, such as the SCARE guidelines, to ensure comprehensive and standardized reporting.
It is unclear why the authors delve into the differential diagnosis of AIN when an AIN diagnosis was ultimately excluded in this case. This point requires clarification or removal to maintain focus.
Please elaborate on why this case is unique and novel compared to the existing literature. Highlight the aspects that make it a significant contribution to the topic
[Reply]: Thank you for your suggestion. Both the title and the entire manuscript have been revised, and changes are highlighted in yellow. Moreover the references were added.
In particular, the authors stress a multidisciplinary approach involving both clinical and surgical setting such as endoscopy in rectal lesion also in emergency setting and that the privacy and the Motivational Interviewing (MI) seems particularly suitable for underserved male youth in this context take in account that Recent studies encourage to enhance knowledge about the health status of lesbian, gay, bisexual, and transgender (LGBT) individuals including sexual health, risk behaviors, kinky practice such as fisting, STIs, HIV rates, and relevant interventions to avoid stigma and promote protective factors for these populations

Reviewer 4 Report
Comments and Suggestions for Authors
1. As a case report, it is based on data from a single patient, which limits its generalizability to the wider population, and there is limited discussion of patient privacy and ethical considerations of sexual behavior communication, which may further emphasize the importance of protecting patient privacy while protecting it. Obtain accurate medical history kept confidential.
2. Although some treatment strategies were discussed, there was no detailed follow-up of patients' long-term outcomes or treatment effects, hampering the evaluation of proposed interventions.
3. Some opinions and conclusions lack sufficient reference, especially regarding complications caused by rare behaviors. Inclusion of more relevant studies would strengthen the argument.
4. Although the report highlights gaps in diagnosis and treatment, it does not provide specific recommendations or strategies, such as improving tools and processes for collecting sexual behavior-related histories.
Comments on the Quality of English LanguageMinor editing of English language required.
Author Response
- As a case report, it is based on data from a single patient, which limits its generalizability to the wider population, and there is limited discussion of patient privacy and ethical considerations of sexual behavior communication, which may further emphasize the importance of protecting patient privacy while protecting it. Obtain accurate medical history kept confidential. Obtain accurate medical history kept confidential.
[Reply]: Thank you for your suggestion.The authors have added the following sentences in abstract and discussion and reference
Abstract:
It highlights the importance of client privacy to reduce stigma and promote access to services, encouraging individuals to seek help.
Discussion: STIs are a significant public health and social burden worldwide, especially in developing countries also relating to privacy and confidentiality in the provision of care for and use of services [28].
Conclusions STIs are a significant public health and social burden worldwide, especially in developing countries also relating to privacy and confidentiality in the provision of care for and use of services [28].
Dapaah, J.M., Senah, K.A. HIV/AIDS clients, privacy and confidentiality; the case of two health centres in the Ashanti Region of Ghana. BMC Med Ethics 17, 41 (2016). https://doi.org/10.1186/s12910-016-0123-3
- Although some treatment strategies were discussed, there was no detailed follow-up of patients' long-term outcomes or treatment effects, hampering the evaluation of proposed interventions.
[Reply]: Thank you for your suggestion. The sentences were added
After several months of attempting to reach the patient regarding missed endoscopy appointments and microbiological tests, we were informed by our Infectious Disease clinician that he had relocated to another country for professional opportunities.
- Some opinions and conclusions lack sufficient reference, especially regarding complications caused by rare behaviors. Inclusion of more relevant studies would strengthen the argument.
[Reply]: Thank you for your suggestion. All references were revised and the authors according to the suggestion of other reviewers have a total of 31 references
- Although the report highlights gaps in diagnosis and treatment, it does not provide specific recommendations or strategies, such as improving tools and processes for collecting sexual behavior related histories.
[Reply]: Thank you for your suggestion. All the conclusion section was revised and outlines in yellow.
Conclusion
Overall, it is an educational instructive case about multidisciplinary approach to patients with rectal lesions according to sex habits to ensuring client privacy to prevent identification and reduce stigmatization. This commitment to privacy encourages continued access to services and motivates potential clients to seek help. There are a few key points that may be suggested 1) Vaccination camping is available and stressful in young adults who are male and have risky sexual behaviors. Still, they are also spreading in the over 45 yrs or in elderly age groups where HPV vaccination is not straightforward unless the patient does not admit their risky behaviors. 2) Sexual behaviors or habits involve the emotional-affective sphere, and the patient is not always available at the first visit to talk about their sexual habits. These practices are not only widespread in the youth population or cases of sexual violence. 3) As the incidence of STDs continues to rise, it would be beneficial for surgical and clinician MD to include data on protection usage in their medical assessments. Furthermore, integrating psychological support alongside medical treatment could improve patient well-being and outcomes. Recently, there has been a focus on promoting autonomy, avoiding labels for client behaviors, and steering clear of aggressive approaches. Motivational Interviewing (MI) seems particularly suitable for underserved male youth in this context. Additionally, it may be more effective to use a harm-reduction approach with male youth who show little or no interest in abstaining from specific behaviors, such as sexual activity [32].4) No interview or questionnaire is available regarding sexual practice in surgical setting and endoscopy because it is difficult to improve the endoscopist doctor-patient relationship. In fact, our patient only at the second counseling reported on his sexual behaviors. 5) In surgical setting the endoscopist is the first doctor who is confronted with the patient who practices anal fisting. The surgeon must be alerted to the possible multiple infections that a subject can acquire during fisting practices that affect not only HIV infection and syphilis but also other pathogens and especially the HPV, NG and CT. Finally, 6) Future research on the role of other non-intercourse receptive anal sexual practices in LGBT subjects, such as fisting, in anal High Risk HPV transmission or reactivation could explored.
Comments on Minor editing of English language required. the Quality of English Language

Round 2
Reviewer 1 Report
Comments and Suggestions for Authors
The review carried out is correct and sufficient. The manuscript can be published in its current form
Author Response
Thank you very much for your help and critical review of the manuscript.

Reviewer 3 Report
Comments and Suggestions for Authors
· The statement, ‘Written informed consent was obtained from the patient's family for publication of this case report and accompanying images,’ indicates that consent was not obtained directly from the patient. In such cases, publication should not proceed, as patient consent is a critical requirement.
· Please ensure adherence to established guidelines for reporting case reports, such as the SCARE guidelines
Author Response
Thank you very much for your request, and we apologize for the initial discordant statement. We have added a new statement in two locations in the manuscript and filled it out with the most updated SCARE guideline checklist, as requested ("
Written informed consent has been obtained initially from the patient to publish this paper; subsequently, since the patient left the city for traveling for work outside of the city boundaries, we also obtained the informed consent from the family. In both events, we assured both patient and family that the case report will not disclose any data to avoid harm to either the patient or the family. Prof. Dr. Rodolico’s statement is uploaded into MDPI platform for transparency. The SCARE guideline checklist was also uploaded to the MDPI platform as requested. Ref.: Sohrabi C, Mathew G, Maria N, Kerwan A, Franchi T, Agha RA; Collaborators. The SCARE 2023 guideline: updating consensus Surgical CAse REport (SCARE) guidelines. Int J Surg. 2023 May 1;109(5):1136-1140. doi: 10.1097/JS9.0000000000000373. PMID: 37013953; PMCID: PMC10389401."). Moreover, the email I received from the colleague with the date and time who wrote documentation in the medical records about the consent forms was also uploaded to MDPI for transparency. We sincerely hope that everything is clarified now.

Reviewer 4 Report
Comments and Suggestions for Authors
The author has modified it according to the suggestion.
Comments on the Quality of English LanguageThe author has modified it according to the suggestion.
Author Response
Thank you very much for your help and critical review of our manuscript.

Round 3
Reviewer 3 Report
Comments and Suggestions for Authors
Thank you for the opportunity to review this revised version of the manuscript.
- Considering that the patient has given consent for publication, the formal consent of the patient’s family is not required. Please modify the statement on patient consent accordingly to minimize potential confounding factors.